# The Role of the Gut Microbiota in Lipid and Lipoprotein Metabolism

**DOI:** 10.3390/jcm8122227

**Published:** 2019-12-17

**Authors:** Yijing Yu, Fitore Raka, Khosrow Adeli

**Affiliations:** 1Molecular Medicine, Research Institute, The Hospital for Sick Children, Toronto, ON M5G 1X8, Canada; yijing.yu@mail.utoronto.ca (Y.Y.); f.raka@mail.utoronto.ca (F.R.); 2Department of Physiology, University of Toronto, Toronto, ON M5S 1A8, Canada; 3Departments of Laboratory Medicine & Pathobiology and Biochemistry, University of Toronto, Toronto, ON M5S 1A8, Canada

**Keywords:** gut microbiota, lipoprotein metabolism, metabolic disorder

## Abstract

Both environmental and genetic factors contribute to relative species abundance and metabolic characteristics of the intestinal microbiota. The intestinal microbiota and accompanying microbial metabolites differ substantially in those who are obese or have other metabolic disorders. Accumulating evidence from germ-free mice and antibiotic-treated animal models suggests that altered intestinal gut microbiota contributes significantly to metabolic disorders involving impaired glucose and lipid metabolism. This review will summarize recent findings on potential mechanisms by which the microbiota affects intestinal lipid and lipoprotein metabolism including microbiota dependent changes in bile acid metabolism which affects bile acid signaling by bile acid receptors FXR and TGR5. Microbiota changes also involve altered short chain fatty acid signaling and influence enteroendocrine cell function including GLP-1/GLP-2-producing L-cells which regulate postprandial lipid metabolism.

## 1. Introduction

The gastrointestinal microbiota represents the largest population of microbial community in the human body. This community contains more than 100 trillion microbial cells inhabiting in the small and large intestine which is estimated to be comparable to the total number of cells in the human body [1]. Additionally, approximately 3.3 million gut microbial genes are detected, which is 150-fold more genes than found in the human genome [2]. Earlier studies have suggested that gut microbiota are involved in several (patho-)physiological processes, including the metabolism of certain nutritional and drug compounds, the development of host immunity, intestinal inflammatory states, and colon cancer development [3,4,5]. Due to the advance of genomic techniques, gut microbiota studies and findings have dramatically expanded in the past decade [6,7]. The two commonly used approaches, 16S or 18S ribosomal RNA (rRNA) sequencing and metagenomic sequencing, have been frequently employed in the microbiota characterization studies. The former approach can detect phylogenetic profiling of microbiota, while the later analyzes all the community DNA within the samples. Moreover, metatranscriptomics or metaproteomics provide transcriptional or protein information for functional studies [8,9]. Although the gut microbiota is composed of more than 1000 species of bacteria, a set of core microbiota harbor in healthy human individuals [8]. Most of the core bacteria belong to five phyla (*Firmicutes*, *Proteobacteria*, *Bacteriodetes*, *Actinobacteria,* and *Veerrucomicrobia*) in humans [10].

The composition and stability of human microbiota exhibit various changes at different stages of life. The first intestinal bacterial colonization occurs during birth, and the infant gut microbiota could be affected by the mode of newborn delivery, the type of feeding, and other factors (e.g., host genetics, antibiotics, prescribed drugs, probiotic and prebiotics supplementation) [11,12]. At three to five years of age, children start to harbor microbiota that are comparable to the ones in adults. After childhood, the gut microbiota become quite stable, although long-term changes can occur in response to the changes in diet and lifestyle, the usage of antibiotics and probiotics, infection, and surgery [13]. The dysbiosis of microbiota has been linked with several human metabolic diseases, such as obesity, diabetes, and nonalcoholic fatty liver disease (NAFLD) [14,15,16,17,18]. In this review, we will present the available evidence implicating gut microbiota in the regulation of lipid and lipoprotein metabolism and potential mechanisms underlying this regulation (Figure 1).

### Role of Microbiota in Metabolic Disorders

Metabolic syndrome is a group of interlinked disorders that includes hypertension, insulin resistance, visceral obesity, low high-density lipoprotein (HDL) cholesterol, and elevated triglycerides [19]. Numerous clinical studies have shown that metabolic syndrome is associated with increased risk of cardiovascular disease and type 2 diabetes [20]. Emerging studies from the last 15 years have implicated gut microbiota as a pathogenic factor affecting many components of the metabolic syndrome. Backhed and colleagues were the first to report a striking difference in body fat content observed between germ-free (GF) mice and conventionally raised mice with the latter having 40% higher body fat content [21]. These observations promoted many future studies exploring the role of gut microbiota in metabolic health. In exploration of the relative abundance of microbial taxa, the gut of obese mice was found to have a 50% reduction in the abundance of *Bacteroidetes* along with a proportional increase in *Firmicutes* [22]. Interestingly, patients with obesity also have a similar alteration in the relative proportion of *Bacteroidetes* and *Firmucutes* and this was shown to improve with a low-calorie diet [23]. 

Alterations in the gut microbiome from high-fat diet (HFD) feeding result in an increased proportion of lipopolysaccharide (LPS)-containing microbiota in the gut which has been linked to increased glycemia and insulinemia [24]. While earlier studies have determined differences in relative abundance of microbial taxa present in the gut of obese mice versus lean mice, recent studies have focused on particular microbial species and their metabolites that mediate the observed effects. For example, imidazole propionate is identified as a microbially produced histidine-derived metabolite that is higher in subjects with type 2 diabetes and was associated with impaired glucose tolerance and insulin signaling when administered to mice [25]. More recently, the tryptophan-derived microbial metabolite indole was found to upregulate the expression of miR-181 in white adipocytes leading to improvements in body weight gain, glucose tolerance, and insulin sensitivity [26]. While changes in gut microbiota occur due to HFD feeding, it is also important to consider that diets high in fat are typically lower in fiber compared to chow controls, which could also contribute to these phenotypes [27]. 

There is also evidence of microbiota dependent protection from metabolic disorders. A study reported that transfer of feces from healthy individuals to individuals with metabolic syndrome via a duodenal catheter alters the fecal microbiota of recipients and is associated with improvements in insulin sensitivity [28]. Studies in mice have revealed specific species to be beneficial for improving metabolic disorders: for example, *Akkermansia muciniphila* treatment has the ability to improve diet-induced obesity, fasting glycemia, and adipose tissue metabolism [29]. While these studies provide insight for the development of therapies that target human gut microbiota for treatment of obesity and its associated metabolic disorders, the intricate process of how specific specifies of bacteria and their metabolites regulate energy metabolism remain unclear. 

## 2. Regulation of Lipoprotein Metabolism by the Gut Microbiome

### 2.1. Introduction of Lipoprotein 

Dietary lipid contributes to around 35% of daily energy resource in humans [30,31]. As lipids are insoluble in water, the human digestive organs have developed a complex system of digestive and absorptive processes by transporting lipids in the form of lipoproteins. Lipoprotein particles are synthesized in the liver and intestine and are composed of lipids (such as phospholipids, cholesterol and triglycerides) and apolipoproteins [31]. Based on size and density, lipoproteins are classified into 5 classes: chylomicron (CM), very low density lipoproteins (VLDL), intermediate density lipoprotein (IDL), low density lipoprotein (LDL), and high density lipoprotein (HDL) particles [32]. Each class of lipoprotein has a specific function in lipid metabolism. CM, as the largest lipoprotein particle, transports dietary triglycerides and cholesterol from the intestine to peripheral tissues, while VLDL are synthetized in the liver to export triglycerides. VLDL particles contain apolipoprotein B 100 (ApoB100) as the main structural apolipoprotein and CM contain ApoB48, a truncated form of ApoB100. Both VLDL and CM are assembled by the microsomal triglyceride transfer protein (MTP) which incorporates lipids into ApoB. LDL particles are the catabolic products of VLDL, while HDL is involved in reverse transport of cholesterol back to the liver [32]. Any deficiency during the lipid digestion process and lipoprotein synthesis cycle could result in dyslipidemia, a key factor in the pathogenesis of metabolic disorders (such as obesity, diabetes, and NAFLD) [33,34,35,36].

The intestine is not only the site for lipid digestion and absorption, but also serves as the major home for microbiota. Gut microbiota has the ability to modulate dietary lipid composition, digestion, and absorption, potentially altering intestinal lipoprotein formation. Abnormal levels of lipoproteins have been observed in GF mice and antibiotic-treated mice. GF mice are found to be resistant to develop an obesity phenotype under HFD feeding [37], partly due to the deficiencies in lipid digestion, absorption, and transport [38]. Recent studies have demonstrated that GF mice exhibit decreased plasma triglyceride and LDL compared to control mice, and this alteration was associated with decreased intestinal absorption and hormone peptide secretion [38]. Furthermore, supplementation of specific strain bacteria (e.g., *Lactobacillus rhamnosus gg*) to conventionally reared mice resulted in body weight gain, increased levels of LDL and cholesterol, and the altered expression of several genes involved in lipid transport [38]. Another study has shown that GF mice have significantly lower levels of fasting triglycerides and VLDL production when compared to conventionally reared mice [37]. To date, there are few systematic analyses of the association between the microbiome composition and lipid profiles in human and animal studies, which will be discussed as follows.

### 2.2. The Association between the Gut Microbiome and Lipid Profile

An early study by Carr et al. showed that supplementation of grain sorghum lipid extract (GSL) significantly improved the HDL/non-HDL cholesterol equilibrium in the Syrian hamster model [39]. A follow-up study by Martinez et al. characterized the interplay among diet, gut microbial ecology, and host metabolism [40]. Using 16S rRNA sequencing, Martinez et al. demonstrated that the abundance of *Bifidobacteria*, which was significantly increased under GSL-supplemented diet feeding, was positively associated with plasma HDL cholesterol levels, while the abundance of *Coriobacteriaceae,* decreased under GSL diet feeding, was positively associated with non-HDL plasma cholesterol levels [40]. 

In humans, Cotillard et al. analyzed the intestinal bacterial genes in 49 obese or overweight individuals by high-throughput sequencing technology [41]. Two groups of humans with high bacterial gene count (HGC) and low bacterial gene count (LGC) were defined in this study. The LGC group exhibited significantly higher frequency of insulin resistance and higher levels of fasting serum triglycerides, as well as higher LDL cholesterol, when compared to the HGC group [41]. Another human study reported by Le Chatelier et al. performed quantitative metagenomics with fecal DNA extracted from 169 obese individuals and 123 nonobese control individuals. They found that the LGC group was associated with elevated serum leptin, triglycerides and free fatty acids (FFA), decreased serum adiponectin, and decreased levels of HDL-cholesterol [42]. 

Fu et al. studied the largest correlation cohort to date, for which fecal samples were collected and the lipid profile determined in 893 participants [43]. Based on α diversity analysis, microbiota operational taxonomic unit (OTU) richness was negatively correlated with body mass index (BMI) and triglycerides, but positively correlated with HDL, while there was no significant correlation between microbial richness and LDL or total cholesterol levels. Furthermore, several microbes were correlated with lipid levels or BMI. For instance, the family of *Clostridiaceae*/*Lachnospiracease* was specifically associated with LDL but not with BMI and other lipoproteins; the genus of *Eggerthella* was associated with increased triglyceride and decreased HDL levels, while the *Butyricimonas* was strongly associated with decreased triglyceride levels [43]. 

In addition, a study of 30 hypercholesterolemia individuals and 27 healthy controls reported differences in commensal bacterial profiles between hypercholesterolemic subjects and controls [44]. In this study, Rebolledo et al. analyzed the fasting whole blood lipid profile and bacterial community profiles by denaturing gradient gel electrophoresis. As expected, hypercholesterolemia subjects had higher levels of serum total cholesterol, triglycerides, and LDL-cholesterol than controls, while they harbored a lower richness and diversity of gut microbiota, thus proposing a potential role of the gut microbiota in the development of hypercholesterolemia [44]. Karlsson et al. performed the metagenomic sequencing with fecal samples collected from 145 European women with normal, impaired, or diabetic glucose control, and reported a negative correlation between serum triglycerides and *Clostridum*, and a positive correlation between HDL and *Clostridium* [45].

Recently, a lifestyle intervention study was performed with a low-saturated-fat and reduced-energy intake diet in 1065 metabolic syndrome patients and healthy subjects [46]. The metabolic syndrome patients had at least three of the following factors: central obesity (waist circumference ≥80 cm in women and ≥90 cm in men), fasting blood glucose ≥ 5.50 mmol/L, triglyceride concentration ≥ 1.65 mmol/L, low levels of HDL cholesterol (<1.00 mmol/L in men and <1.25 mmol/L in women), and systolic and/or diastolic blood pressure ≥ 130/85 mm Hg. Positive association between the body mass index and the gut microbiota dysbiosis, and negative association between insulin resistance and HDL-C concentration were found in metabolic syndrome patients. A short period (15 days) of diet intervention decreased the level of serum triglycerides, while the longer-term (75 days) intervention led to a significant reduction in VLDL, CM, LDL, small HDL particle, and improved glucose tolerance in 44.8% of metabolic syndrome patients. Moreover, the lifestyle intervention ameliorated gut dysbiosis, resulting in a reduction in the P/ B ratio and an increase in the abundance of *Akkermansia muciniphila* and *Faecalibacterium prausnitzii* [46].

### 2.3. Evidence from Other Studies

Atherosclerosis-prone apolipoprotein E-deficient (ApoE-/-) mice exhibit delayed lipoprotein clearance and consequently develop dyslipoproteinemia [47]. Antibiotic (i.e., ampicillin) treatment was found to transiently improve glucose tolerance, lower plasma LDL and VLDL cholesterol levels, and reduce aortic atherosclerotic lesion area in Apo E-/- mice, suggesting that alteration of gut microbiota can attenuate dyslipoprotienemia in this model [48]. Additionally, a few studies have reported that gut microbiota could regulate the production of apolipoproteins and MTP [49,50]. Sato et al. found that rats receiving four days of antibiotics (streptomycin and penicillin) had reduced lymphatic transport of triglycerides and phospholipids, and decreased levels of mucosal apolipoproteins B, A-I, and A-IV, suggesting that gut microbiome promotes lipid absorption, probably by regulating intestinal production of apolipoproteins and secretion of CM [51]. Jin et al. also reported that six weeks of antibiotic (penicillin G (Pen G) and erythromycin (Ery)) treatment in mice resulted in increased hepatic lipid accumulation and higher circulating triglyceride levels and expression of key genes involved in FFA synthesis, triglyceride synthesis, and transport including MTP. [49]. 

A recent study in hamsters suggested that soybean sterols could alleviate HFD-diet-associated liver pathology and abnormal cholesterol metabolism by altering both host gene expression and gut microbiota composition [52]. Specifically, compared to the HFD vehicle group, the supplementation of soybean sterols significantly reduced the levels of triglycerides, total cholesterol, and non-HDL-cholesterol, increased the expression of cholesterol absorption/sterol excretion-related genes (e.g., NPC1L1, ABCG5, and ABCG8), and decreased the expression of cholesterol biosynthesis-related genes (e.g., SREBP2 and LDL-R). The addition of soybean sterols significantly increased the abundance of *Lactobacillus*, *Oscillospira*, and *Akkermansia*, and also the levels of all six short-chain fatty acids in the fecal content. In addition, a few studies have shown plant sterol esters could decrease plasma LDL cholesterol levels in humans [53,54,55]. A follow-up study revealed the link between gut bacteria and the cholesterol-lowering ability of plant sterol esters [56]. In this study, 13 healthy people with elevated plasma LDL cholesterol were administered with Stearate-enriched plant sterol esters for four weeks. This supplementation resulted in microbiota population change according to 16s rRNA pyrosequencing, and also correlated significantly to changes in serum cholesterol concentrations [56]. 

## 3. Mechanisms of the Role of Microbiota in Host Lipid Metabolism

### 3.1. Bile Acids

Bile acids are the major functional component of bile, which are synthesized in the liver and stored in the gall bladder, and further released in the intestine. One of the major functional roles of bile acids is to emulsify dietary fat [57]. However, bile acids are also known to act as important signaling molecules involved in metabolic pathways (i.e., glucose and lipid metabolism) [58] via interacting with various host bile acid receptors, such as nuclear receptors farnesoid-X-receptor (FXR) and cell membrane receptor Takeda G protein-coupled receptor 5 (TGR5) [58].

Based on recent studies of microbiota depletion or GF animal models, gut microbiota was found to be a key player in the regulation of lipid metabolism via bile acid receptor signaling. Many studies have shown that the remarkable change in gut microbiota induced by antibiotics treatment correlates with decreased intestinal FXR signaling as well as modified bile acid composition [59,60,61,62]. When comparing GF and conventionally raised mice, dramatically different composition of the bile acid pool and expression profile of genes involved in bile acid synthesis, conjugation, and reabsorption were observed [62]. Kuno et al. found a five-day antibiotics (vancomycin and polymyxin B) treatment decreased amounts of secondary bile acid-producing bacteria in feces and reduced the levels of secondary bile acid [lithocholic acid (LCA) and deoxycholic acid (DCA)] [63]. Another hamster study reported that four weeks of an antibiotics cocktail alleviated HFD-induced hepatic steatosis and glucose intolerance, which correlated with the modulation of gut microbiota, bile acids, and FXR signaling [64]. However, Reijnders et al. reported that a seven-day antibiotic (amoxicillin or vancomycin) treatment did not have any clinical impact on host metabolism in obese humans, although changes in microbiota, short-chain fatty acids levels, and bile acid levels were observed [65]. 

Bile acids are originally produced from cholesterol in hepatocytes as conjugated and primary bile acid (cholic acid (CA) and chenodeoxycholic acid (CDCA)). Specific enzymes (i.e., bile salt hydrolase (BSH), bile acid-inducible (BAI) enzymes) deconjugate and dehydroxylate these primary bile acids to generate unconjugated and secondary bile acids (lithocholic acid (LCA), deoxycholic acid (DCA)) [57]. Recent studies have shown that gut microbiota control the homeostasis of bile acids, thus impacting various host pathophysiological processes [58,66]. BSH, which was discovered in 1995, is one of the well-known bacterial bile acid deconjugating enzymes [67,68]. BSH has been detected in various commensal gut microbial species belonging to both gram-positive and gram-negative bacteria, including *Lactobacillus*, *Bifidobacterium*, *Enterococcus*, *Clostridium spp,* and *Bacteroides spp*. A functional metagenomics study revealed that functional BSH is abundantly present in the gut and enriched in the human gut microbiome [69].

In addition, it has been proposed that there is bidirectional interaction between gut microbiota and host bile acid profiles. Changes of the gut microbiota could alter the expression level of BSH and subsequently result in changes of bile acid pool, as evidenced by antibiotics treatment or GF animal studies [62,70]. In addition, bile acids are known to be toxic to bacteria and affect the growth rate of certain bacteria. In vivo studies suggest that bile acids alter microbiota composition and are associated with host metabolic changes. Islam et al. found that supplementation of primary bile acid (i.e., cholic acid) in the diet could alter plylum-level composition of gut microbiota, and similar changes were found in HFD studies [71]. In another study, GF and conventional mice were fed with different diets (control diet, control diet plus primary bile acid CA, and CDCA and HFD) for eight weeks [72]. Enhanced fat mass accumulation and metabolic changes were observed in conventional mice, but not in GF mice. Besides the impaired glucose metabolism, elevated amounts of hepatic triglycerides, cholesteryl esters, and monounsaturated fatty acids were shown in the bile acids treatment group [72]. Additionally, a shift of gut microbiota communities were found in bile acids-fed mice, including increased abundance of *Desulfovibrionaceae*, *Clostridium lactatifermentans,* and *Flintibacter butyricus,* and decreased abundance of *Lachnospiraceae* [72]. Moreover, according to metatranscriptomic analysis, changes of genes involved in lipid and amino acid metabolism were observed in this study [72]. In addition, the bile acid profiles and microbiota were found to be altered in certain disease conditions, such as obesity and type 2 diabetes (T2D), which also indicates a link between bile acid and gut microbiota [66].

#### Role of Bile Acid Signaling through FXR and TGR5

Recent studies have revealed that alteration of gut microbiota could not only affect the bile acid pool, but also influence the bile acid receptor signaling (i.e., FXR and TGR5). FXR is a nuclear receptor which is expressed in the liver, stomach, duodenum, jejunum, ileum, colon, white adipose tissues, kidney, gall bladder, heart, kidney, and adrenal cortex adrenal gland [73]. Bile acids act as ligands to bind and activate FXR, with CDCA and its conjugated forms being the most potent natural agonists (CDCA >LCA >DCA >CA) [74]. FXR has been reported to be involved in glucose homeostasis, energy expenditure, and lipid metabolism. Watanabe et al. has shown that CA could prevent hepatic triglyceride accumulation, decrease VLDL secretion, and lower serum triglyceride in a type 2 diabetes mouse model of hypertriglyceridemia [75]. Furthermore, the protective effects were due to an increase in short heterodimer partner (SHP) following FXR activation, resulting in a subsequent reduction of hepatic sterol regulatory element binding protein-1c (SREBP-1c) expression and other lipogenic genes [75]. An earlier study has shown that CDCA could activate FXR signaling and lead to decreased expression of MTP and apoB, thereby lowering lipoprotein production in HepG2 cell culture [76]. Using Syrian golden hamsters, Bilz et al. have also shown that activation of FXR by CDCA in a high-fructose diet could reduce VLDL triglycerides, VLDL, and IDL/LDL cholesterol levels, mainly by decreasing de novo lipogenesis and hepatic secretion of triglyceride-rich lipoproteins [77]. FXR stimulatory bacteria strains that were selected based on a luciferase assay with the FXR reporter were able to improve HFD-induced obesity phenotype in mice [78]. Controversial results were, however, reported from the study on human non-alcoholic fatty liver disease, showing that decreased activation of liver FXR was found in these patients and associated with increased expression of liver X receptor, SREBP-1c, and hepatic steatosis [79]. However, another study showed FXR signaling was activated in obese human intestines, and intestine-selective FXR inhibitor improved obesity and metabolic syndrome in HFD-induced obese mice [80].

TGR5 is a transmembrane G-protein-coupled receptor and widely expressed among various tissues, including gallbladder, small intestine, colon, liver, brain, skeletal muscle, heart, and the enteric nervous system. Bile acids can bind to TGR5 and act as potent agonists (with rank of LCA> DCA >CDCA> CA) [81]. Similar to FXR, TGR5 activity can regulate glucose and lipid homeostasis. Several in vivo studies in mice have suggested that TGR5 activation by synthetic agonists could improve insulin and glucose tolerance as well as liver and pancreatic function [82], decrease plasma triglycerides and LDL-cholesterol [83], and ameliorate hepatic steatosis [84]. However, controversial results have been reported with a TGR5 KO mice model, as TGR5 KO female mice gained much more body weight and accumulated more fat mass under HFD feeding, while TGR5 knockout male mice did not show significant difference in body weight gain or fat mass accumulation when compared to the wild-type control [85]. Another study reported that TGR5 KO mice have decreased fasting-induced hepatic lipid accumulation, an increased hepatic fatty acid oxidation rate, and decreased hepatic fatty acid uptake [86].

Additionally, activation of FXR and/or TGR5 was reported to increase glucagon-like peptide-1 (GLP-1) secretion, a hormone peptide which contributes to maintaining the metabolic homeostasis. Pathak et al. has shown that both FXR and TGR5 agonists could increase glucose-induced GLP-1 secretion [87]. In addition, activation of TGR5 increased intracellular cAMP, which further enhanced GLP-1 secretion [82]. Additionally, a TGR5 agonist was able to induce GLP-1 secretion in human studies [88], while glucose-induced GLP-1 secretion was reduced in FXR or TGR5 KO mice compared to control mice [89]. The role of GLP-1 in lipid and lipoprotein metabolism will be discussed below.

### 3.2. Short Chain Fatty Acids (SCFAs)

SCFAs are the fermentation products of non-digestible carbohydrates by gut microbiota. Gut microbiota have enzymes that the host lacks to break down these carbohydrates. The undigested dietary fibers as well as proteins that “escape” digestion and absorption are sequentially fermented in the small and large intestines by gut bacteria [90]. SCFAs can be divided into five groups: formic acid (C1), acetic acid (C2), propionic acid (C3), butyric acid (C4), and valeric acid (C5). The majority of gut SCFAs are acetate, propionate, and butyrate, which constitute more than 95% of the SCFA content [91]. After production, SCFA can be absorbed in the cecum, colon, and rectum, and then enter the mesenteric vein, drain into the portal vein, and finally into circulation [92]. After entering the circulation, SCFA can affect the metabolism of several peripheral tissues, including liver, adipose tissue, and skeletal muscle [93]. The advent of advanced genomic sequencing approaches has yielded great progress in identifying the bacteria responsible for SCFA production. Among the three major gut SCFAs, propionate and butyrate production are detected within a few conserved and substrate specific bacteria, while acetate production are widely distributed among bacterial groups. For butyrate production, a small number of gut bacteria, including *Faecalibacterium prausnitzii*, *Eubacterium rectale*, *Eubacterium hallii,* and *Ruminococcus bromii* are the major responsible strains [94]. Additionally, a few studies studying specific dietary ingredients have associated SCFA production with certain bacterial species. For instance, resistant starch greatly contributed to butyrate production in the colon predominantly by *Ruminococcus bromii* [95], while mucin was fermented into propionate mainly by *Akkermansia municiphilla* [96].

SCFAs play key roles in various physiological processes, such as regulation of energy intake, energy harvesting, glucose metabolism, lipid metabolism, adipogenesis, and immune responses, as well as pathophysiology of obesity and related metabolic disorders. Both animal and human studies support the relationship between the gut microbiota, SCFA, and metabolic disorders (i.e., obesity, T2D, NAFLD). Obese mice and humans show a changed ratio of *Firmicutes* to *Bacteroidetes* and increased fecal concentration of SCFA [97,98,99,100]. The obese phenotype of Toll like receptor 5 KO mice were also associated with gut dysbiosis and enhanced cecal SCFA production [101]. In humans, the butyrate-producing bacteria have been associated with the beneficial metabolic effects of the fecal transplantation from lean donors to obese patients [28], while a lower capacity to ferment complex carbohydrates was found in the microbiota of obesity patients compared to lean controls [102]. In the liver, acetate and butyrate are the major substrates for de novo lipogenesis, as well as substrates in cholesterolgenesis, while propionate appears to be an effective inhibitor of both processes [103]. Under physiological conditions, SCFAs such as acetate and propionate were found to inhibit adipose tissue lipolysis [104], resulting in reduced FFA flux from the adipose tissue to liver, but an opposite phenotype was observed in fatty liver disease. A previous human study also reported that propionate supplementation in the diet was able to prevent body weight gain and liver fat accumulation in NAFLD patients [105]. 

Many studies have suggested that SCFA are related to the release of gut-derived satiety hormones, in particular GLP-1 and peptide YY (PYY). Based on in vitro studies using intestinal cell lines from both rodents and humans, SCFAs can stimulate the secretion of PYY and GLP-1 from L-cells, which were dependent on the SCFA receptors GPR41 and GPR43 [106,107,108,109,110]. Furthermore, evidence from in vivo studies has shown that carbohydrate feeding (i.e., oligofructose, inulin, and resistant starch) can increase PYY and GLP-1 secretion in rodents [111,112,113]. Moreover, human studies with either supplementation of SCFA or fermentable polysaccharides (i.e., oligofructose) have shown that weight loss, improved hepatic lipid content, and glucose metabolism were associated with enhanced PYY and GLP-1 release and secretion [105,114,115,116]. However, it warrants further studies to investigate the effects of physiologically relevant SCFA mixtures on the secretion of GLP-1 and PYY and their metabolic consequences in humans.

### 3.3. Enteroendocrine Cell Regulation of Hormone Secretion

The intestine harbors specialized cells called enteroendocrine cells that secrete hormones in response to nutrients and microbial metabolites. In turn, these gut hormones can regulate nutrient metabolism and feeding behavior [117]. Classification of enteroendocrine cells was originally based on expression of one or two hormones, however, transcriptional profiling has determined that these cells can express multiple hormones, making them less distinct than previously recognized [118]. There is accumulating evidence that microbiota can regulate enteroendocrine cell development and function. Intestinal enteroendocrine cells like absorptive enterocytes have an apical membrane facing the gut lumen which allows for interaction with microbes and their metabolites. 

#### 3.3.1. Microbial Regulation of Enteroendocrine L-Cells

Two gut hormones discovered by our lab as being important regulators of intestinal lipid metabolism are GLP-1 and GLP-2, co-secreted in equimolar amounts by enteroendocrine L-cells. Microbiota-derived products such as SCFAs have been shown to stimulate GLP-1 secretion via activation of the G-protein-coupled receptor FFAR2. Microbiota can also influence the L-cell transcriptome. Using transgenic GLU-Venus mice driving expression of yellow fluorescent protein under the proglucagon promoter, Arora et al. found that genes related to vesicle organization and synaptic vesicle cycle such as synaptophysin, synaptotgamins, Rab, and SNAP proteins were downregulated in L-cells from conventionally raised GLU-Venus mice compared to GF GLU-Venus mice, suggesting GF mice have enhanced hormone secretion [119]. They also observed that L-cells from conventionally raised GLU-Venus mice have upregulated expression of nutrient sensing receptors and downregulated expression of the bile acid receptor TGR5 compared to GF GLU-Venus mice [119]. Studies over the past few years have revealed the link between gut microbiota and GLP-1 signaling. Evidence has shown a higher plasma GLP-1 level observed in GF mice and antibiotic treated animals compared to controls [38,120]. Importantly, activation of TGR5 has been shown to increase GLP-1 secretion, and enhanced TGR5 expression in GF mice could contribute to elevated GLP-1 levels found in GF mice [121]. A few studies have also shown that prebiotics administration results in increased level of GLP-1 in the circulation of rats and humans [111,114,122]. Additionally, the beneficial effects of prebiotics in diabetes and obesity models were dependent on a functional GLP-1, as the protective effects were not observed in GLP-1R KO mice [123]. The fermentation products of gut microbiota, such as SCFAs and bile acids, may underline the mechanisms for the crosstalk between microbiota and GLP-1 since microbiota metabolites are involved in stimulating the secretion of certain gut peptide hormones, including PYY, GLP-1, and GLP-2. SCFAs are able to bind to and activate selected G-protein-coupled receptors in different tissues to regulate peptide hormone secretion (e.g., activation of GPR41 and 43 to release GLP-1 secretion) [106,107,108,109,110]. Thus, microbiota and their metabolites have the capacity to regulate enteroendocrine L-cell function leading to altered hormone release. The roles of GLP-1 and GLP-2 in regulating lipoprotein metabolism will be discussed below.

##### GLP-1

As an incretin hormone, GLP-1 is well known to induce insulin secretion and subsequently decrease the circulating glucose levels [124,125]. However, GLP-1 also regulates appetite, body weight loss, and lipid metabolism [126,127,128]. In addition to decreasing adipose lipolysis [129], GLP-1 is involved in regulation of intestinal and liver lipid/lipoprotein metabolism [128]. The role of GLP-1 in modulating intestinal lipoprotein production has been extensively studied in both animal models as well as humans [130]. Administration of GLP-1 receptor agonist exendin-4 or dipeptidyl peptidase 4 (DPP-4; as a GLP-1 degradation enzyme) inhibitor in hamsters both significantly reduce intestinal lipoprotein production and plasma triglycerides and cholesterol within the triglyceride-rich lipoprotein fraction [130]. Conversely, blocking GLP-1 receptor signaling by infusion of the GLP-1 receptor antagonist exendin 9-39 or using GLP-1 receptor knockout mice enhanced apoB48-containing lipoprotein secretion [130]. Similar effects were shown in a rat model, while administration of native GLP-1 decreased intestinal fat absorption and production of apoB and apoAIV [131]. Administration of DDP-4 inhibitor in T2D patients exhibited benefit effects on postprandial levels of triglycerides (TG), apoB48, CM triglycerides [132] and hepatic triglycerides [133,134]. Antidiabetic drug exenatide (53% GLP-1 homology) reduced postprandial triglycerides and apoB48 in T2D patients [135]. Overall, GLP-1 activity has been shown to exert very beneficial effects on lipid and lipoprotein homeostasis.

##### GLP-2

Alongside GLP-1, its sister peptide GLP-2 is co-secreted from L-cells upon luminal lipid sensing [136]. GLP-2 plays an important role in increasing the absorption of carbohydrates and amino acids from the intestinal lumen by increasing nutrient transporter expression [137,138,139]. Although secreted in equimolar amounts, these hormones have opposing roles in regulation of postprandial lipids. In a hamster model and in healthy humans, GLP-2 was found to raise postprandial CM production­­ and plasma triglycerides [140,141]. A GLP-2 mediated increase in CM production in hamsters was associated with increased apical expression of the fatty acid translocase CD36 which is involved in fatty acid uptake [142]. Interestingly, coadministration of GLP-1 and GLP-2 for 30 minutes in hamsters results in a GLP-2 dominant effect and an increase in CM secretion [143]. Similarly, in an insulin resistant model, the stimulatory actions of GLP-2 appear to override the inhibitory effects of GLP-1 [143]. Considering that the half-life of GLP-2 is longer than GLP-1, increasing L-cell hormone secretion may result in a GLP-2 dominant effect, leading to excess accumulation of atherogenic CM particles. Thus, microbial regulation of enteroendocrine L-cells can have profound implications for lipoprotein metabolism.

#### 3.3.2. Microbial Regulation of Enterochromaffin Cells

Microbial regulation has also been observed for other enteroendocrine cells. Indigenous spore-forming microbes from colons of specific pathogen free (SPF) mice were found to increase colonic and blood serotonin levels and these spore forming microbes were associated with a particular set of metabolites that positively correlated with serotonin levels [144]. In line with this, mice inoculated with spore-forming *Clostridium ramosum* have an increased number of serotonin-producing cells known as enterochromaffin cells [145]. This was shown to occur through increases in transcription factors regulating enterochromaffin cell development. As a result, mice inoculated with C. *ramosum* had higher levels of serotonin and gained more weight on a HFD. This is consistent with emerging studies proposing a role for gut-derived serotonin in obesity [146,147,148]. Interestingly, gut-derived serotonin has been shown to positively regulate liver lipid metabolism [149,150,151]. Using tryptophan hydroxylase 1 knockout mice which lack the enzyme for synthesizing serotonin in the gut, Choi et al. found that these mice are protected from HFD-induced fatty liver [149]. Under HFD feeding, these mice showed decreased expression of genes involved in lipogenesis but no differences in apoB or MTP expression. This would suggest that that serotonin does not affect VLDL secretion, however no experiments besides gene expression were done to directly measure VLDL secretion in this model. Considering the close proximity of enterochromaffin cells to absorptive enterocytes, it is possible that serotonin could modulate aspects of intestinal lipid metabolism. Studies with C. *ramosum* found that colonization by this serotonin-enhancing bacterium increased expression of lipid transporters such as CD36 and FATP4 in the colon [145]. Furthermore, serotonin treatment of Caco-2 cells also increased CD36 and FATP4 protein levels [145]. Given that the colon is generally not a major site of lipid absorption, the significance of these findings in serotonin-mediated lipid absorption are unclear, however the role of serotonin in small intestine lipid absorption and metabolism remains to be elucidated. 

### 3.4. Gut Barrier Function

Barrier function at the intestinal mucosal interface enables absorption of water and essential nutrients, but also protects against ingested or endogenous toxins. The intestinal epithelial layer consists of absorptive enterocytes, goblet cells, enteroendocrine cells, tuft cells, and Paneth cells. Substances can pass through this barrier via a transcellular route involving selective transporters or simple diffusion. Alternatively, they can pass through in the space between epithelial cells, a process known as paracellular diffusion. Some residing bacteria produce LPS which activate Toll-like receptor 4 (TLR4) [152]. LPS is restricted within the gut lumen, however, when intestinal barrier function is weakened, LPS can enter the blood circulation and trigger systemic inflammation [24]. Lebrun et al. have demonstrated that when gut barrier function is compromised as is with dextran sodium sulfate treatment or via ischemia/reperfusion in mice, LPS stimulates a rapid and robust rise in GLP-1 [153]. The authors attributed these findings to mechanisms involving enhanced LPS access to TLR4 receptors located on GLP-1-secreting enteroendocrine cells when gut barrier function was altered. Thus, gut barrier function can have important implications for enteroendocrine cell responses to microbial metabolites. 

Additionally, LPS can be internalized by enterocytes through a transcellular route involving a TLR4-dependant mechanism and can be incorporated into CM [154]. LPS absorption from the gut was found to be associated with fat absorption since plasma LPS concentrations increased by 50% after ingestion of a high-fat meal [155]. While increased LPS into the circulation via enhanced gut permeability is known to be associated with systemic inflammation [24], the role of LPS contained within CM is unknown. In states of insulin resistance, postprandial overproduction of CM particles is observed and leads to an increase in atherogenic CM remnant particles [156,157,158,159]. Interestingly, long-chain triglyceride ingestion in mice led to an increase in plasma LPS which was mainly associated with the CM remnant fraction [154]. LPS contained within CM remnants could contribute to their proatherogenic nature since elevated concentrations of circulating LPS correlate well with an increased atherosclerosis risk [160]. LPS injection accelerates plaque formation in both mice and rabbits [161,162] and apoE-/- mice with a genetic deletion of the LPS receptor TLR4 have significantly reduced plaque formation [163,164]. Whether LPS alone or LPS associated with CM remnants is responsible for these observations remains unknown.

### 3.5. Other Microbiota Metabolites

Gut microbiota are known to provide a small proportion of amino acids to host by de novo sythesis. Hoyles et al. reported a positive association of hepatic steatosis with gut microbial amino acid metabolism based on the study of 56 morbidly obese, weight-stable, nondiabetic women [165]. Interestingly, animo acid phenylacetate (PAA) was strongly correlated with steatosis, which was found to be mainly produced by *bacteroides spp* in humans. Both fecal microbiota transplants and chronic phenylacetate treatment revealed that phenylacetate induced liver steatosis. Serum PAA may thus be a useful biomarker and a potential therapeutic target for treatment of hepatic steatosis [166].

## 4. Concluding Remarks

A key role for gut microbiota in modulating metabolic health is now widely acknowledged, but the underlying mechanisms are not clearly understood. Of particular interest are the links between dietary nutrients, intestinal flora, and the regulation of carbohydrate and lipid metabolism. Although modulation of lipid and lipoprotein homeostasis by gut microbiota is largely unexplored, emerging evidence suggest important roles in regulation of dietary fat absorption and postprandial lipid metabolism. Evidence also suggests a potential role in regulating hepatic lipid accumulation and development of hepatic steatosis. These potential links appear to be mediated by microbiota-mediated regulation of intestinal enteroendocrine system and secretion of specific gut peptides. Complex interactions between bacterial populations in the gut and the intricate network of enteric neurons as well as gut immune system are also likely to play important roles in modulating nutrient metabolism and metabolic health. However, no direct experimental evidence currently exists, and new research efforts are needed to explore these links and elucidate the underlying mechanisms.

## Figures and Tables

**Figure 1 jcm-08-02227-f001:**
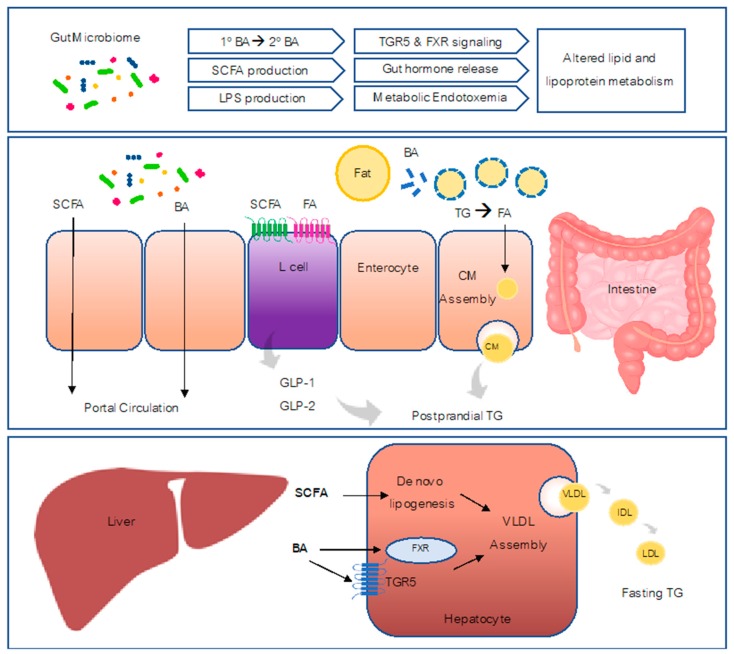
Mechanisms of the role of microbiota in host lipoprotein metabolism. The intestine and liver are two major organs involved in lipid and lipoprotein metabolism. In the intestine, BA emulsify fat into smaller fat particles which allows lipases to breakdown triglycerides into fatty acids. Fatty acids can then be absorbed and be used as substrates for CM assembly. CM production contributes to postprandial TG levels. Microbiota within intestine generates SCFA, secondary BA and LPS. SCFA and FA can activate receptors on enteroendocrine L-cells to release GLP-1/GLP-2 which regulate postprandial CM production. SCFA and BA are absorbed and enter the portal circulation. In the liver, SCFA can act as substrates for de novo lipogenesis and contribute to VLDL production. BA can activate FXR and TGR5 signaling pathways to regulate VLDL assembly. VLDL and its products IDL and LDL contribute to fasting TG levels. BA, bile acids; CM, chylomicron; FXR, farnesoid X receptor; FA, fatty acids; GLP-1/2, Glucagon-like peptide 1/2; IDL, intermediate density lipoprotein; LPS, lipopolysaccharide; LDL, low density lipoprotein; SCFA, short-chain fatty acid; TGR5, G-protein-coupled bile acid receptor; TG, triglyceride; VLDL, very low density lipoproteins.

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
