# Peer review of "The Role of the Gut Microbiota in Lipid and Lipoprotein Metabolism"

_jcm, 2019, doi:10.3390/jcm8122227_

Round 1

Reviewer 1 Report

I really appreciate my lecture of this review on gut microbiota and lipid metabolism. I find it quite novel and original as it does bring new information and a new angle of view (link with lipid). I have only moderately important issues to improve the review.

1) line 27: should update the information about the number of bacteria. Estimates are now closer to 1:1 ratio.

2) line 66: Should insist on the fact that these diets have drastically less fibre and this is probably the most important on gut microbiota.

3) Line 44: Should indicate drug prescription as these molecules, more than the metabolic state, are known to influence gut microbiota

4) line 200: the role of bile acid is not to digest lipids!

5) lines 248 to 290: Should rethink this section. The authors deviate from the original goal of the review and discuss bile acid and receptors rather than microbiota. It is not necessary, or not that long.

6) lines 333 to 344: Should reformulate to help the reading

7) lines 383 to 403: Same as point 5. There is no relation with microbiota and reviewing GLPs is not the scope of the review even if the author is the expert!

In conclusion, the review is very good but could be improved by removing some unnecessary paragraphs and sentences regarding the scope of the review (gut microbiota and lipid metabolism).   

Reviewer 2 Report

Include some figures explaining lipoprotein metabolism in gut.

My main concern is this review article does not have any figures, it would be really easy for readers to understand if its have any figures. Also include some appropriate references, in many places appropriate references are not cited.
